# NGS Analysis of Liquid Biopsy (LB) and Formalin-Fixed Paraffin-Embedded (FFPE) Melanoma Samples Using Oncomine™ Pan-Cancer Cell-Free Assay

**DOI:** 10.3390/genes12071080

**Published:** 2021-07-16

**Authors:** Magdalena Olbryt, Marcin Rajczykowski, Wiesław Bal, Anna Fiszer-Kierzkowska, Alexander Jorge Cortez, Magdalena Mazur, Rafał Suwiński, Wiesława Widłak

**Affiliations:** 1Center for Translational Research and Molecular Biology of Cancer, Maria Sklodowska-Curie National Research Institute of Oncology Gliwice Branch, 44-102 Gliwice, Poland; wieslawa.widlak@io.gliwice.pl; 2Radiotherapy and Chemotherapy Clinic and Teaching Hospital, Maria Sklodowska-Curie National Research Institute of Oncology Gliwice Branch, 44-102 Gliwice, Poland; marcin.rajczykowski@io.gliwice.pl (M.R.); rafal.suwinski@io.gliwice.pl (R.S.); 3Chemotherapy Day Unit, Maria Sklodowska-Curie National Research Institute of Oncology Gliwice Branch, 44-102 Gliwice, Poland; wieslaw.bal@io.gliwice.pl; 4Department of Genetic and Molecular Diagnostics of Cancer, Maria Sklodowska-Curie National Research Institute of Oncology Gliwice Branch, 44-102 Gliwice, Poland; anna.fiszer-kierzkowska@io.gliwice.pl (A.F.-K.); magdalena.mazur@io.gliwice.pl (M.M.); 5Department of Biostatistics and Bioinformatics, Maria Sklodowska-Curie National Research Institute of Oncology Gliwice Branch, 44-102 Gliwice, Poland; alexander.cortez@io.gliwice.pl

**Keywords:** melanoma, liquid biopsy, targeted next-generation sequencing

## Abstract

Next-generation sequencing (NGS) in liquid biopsies may contribute to the diagnosis, monitoring, and personalized therapy of cancer through the real-time detection of a tumor’s genetic profile. There are a few NGS platforms offering high-sensitivity sequencing of cell-free DNA (cfDNA) samples. The aim of this study was to evaluate the Ion AmpliSeq HD Technology for targeted sequencing of tumor and liquid biopsy samples from patients with fourth-stage melanoma. Sequencing of 30 samples (FFPE tumor and liquid biopsy) derived from 14 patients using the Oncomine™ Pan-Cancer Cell-Free Assay was performed. The analysis revealed high concordance between the qPCR and NGS results of the BRAF mutation in FFPE samples (91%), as well as between the FFPE and liquid biopsy samples (91%). The plasma-tumor concordance of the non-BRAF mutations was 28%. A total of 17 pathogenic variants in 14 genes (from 52-gene panel), including *TP53*, *CTNNB1*, *CCND1*, *MET*, *MAP2K1*, and *GNAS*, were identified, with the *CTNNB1*^S45F^ variant being the most frequent. A positive correlation between the LDH level and cfDNA concentration as well as negative correlation between the LDH level and time to progression was confirmed in a 22-patient cohort. The analysis showed both the potential and limitations of liquid biopsy genetic profiling using HD technology and the Ion Torrent platform.

## 1. Introduction 

According to the NCI definition, liquid biopsy is a test done on a sample of blood to look for cancer cells from a tumor that are circulating in the blood or for pieces of DNA from tumor cells that are released into the blood (circulating tumor DNA, ctDNA). For the last years, liquid biopsy has been extensively explored for cancer diagnosis, prognosis, disease monitoring, and therapy selection in various cancers, including melanoma [1,2], 2020). Liquid biopsy provides representative tumor material and is easily accessible; therefore, it seems to be a convenient tool especially for monitoring disease progression, response to therapy, and tumor evolution [3]. On the other hand, NGS analysis enables the identification of many genetic variants in many samples in a short time, with the sensitivity reaching 0.1% of the variant allele frequency (VAF) for some NGS technologies [4]. The combination of these two tools (NGS of ctDNA) enables genetic profiling of the tumor, which may be used for monitoring tumor resistance and evolution as well as the selection of therapeutic targets [5]. 

Melanoma is one of the cancers for which genetic analysis of liquid biopsy may revolutionize patient treatment. Targeted therapy and immunotherapy have brought enormous progress in the treatment of advanced melanoma, with elongation of the median survival time from nine months to an undefined timeframe, and with a subset of patients having long-term tumor control [6]. However, despite this unquestionable improvement, approximately 10–20% do not benefit from targeted therapy at all, and most of the responding patients experience progression within 7–13 months, depending on the therapy: monotherapy (anti-BRAF) vs. combination (anti-BRAF + anti-MEK) [7]. In the case of immunotherapy, 20–60% objective responses were observed, depending on the treatment modality [8], and approximately 7% of patients experienced pseudo-progression [9]. Identification of predictive markers together with better monitoring of the disease could improve personalized treatment of melanoma patients. There are three types of diagnostic practices that can be improved using liquid biopsy genetic tests: monitoring of treatment efficacy, prediction of treatment response, and selection of therapeutic targets. For these purposes, various techniques and tests are being evaluated in various cancer types [3,10]. Some of them have already been approved by the FDA, including the first test, which combines NGS and liquid biopsy—accepted for tumor profiling across all solid cancers (Liquid Biopsy Next-Generation Sequencing Companion Diagnostic Test, The Guardant360 CDx assay). For research purposes, however, tests tailored to the particular aim of the study are required [11,12]. One of the NGS platforms on the market is the Ion Torrent system (Thermo Fisher Scientific), which, using HD technology, enables the detection of rare genetic variants (VAF > 0.1%) in liquid biopsy material. The main aim of our study was to evaluate the applicability of this platform and technology for next-generation sequencing of tumor and liquid biopsy melanoma samples using Oncomine™ Pan-Cancer Cell-Free Assay. We intended to assess the technology performance and the workflow of the Ion Torrent system as a tool for future research into the use of genetic profiling in melanoma diagnosis, monitoring, and treatment. Additionally, on our patient cohort, we have evaluated the correlation between the following parameters: concentration of cfDNA, LDH level (typically used to monitor disease burden), and time to progression (TTP).

## 2. Materials and Methods 

### 2.1. Patients and Treatments 

Twenty-two patients with fourth-stage melanoma according to the *TNM Classification of Malignant Tumors*, 8th edition, treated for the first time with either targeted therapy or immunotherapy in the Maria Sklodowska-Curie National Research Institute of Oncology (NRIO), Gliwice Branch, were included in this study. Before systemic treatment, ten patients underwent surgery and two of them received adjuvant radiotherapy. Seven patients were treated with radiotherapy (stereotactic body irradiation and/or whole-brain radiotherapy) before systemic treatment because of brain metastases. Two of them had also brain metastasis surgery before radiotherapy. Patients were in good clinical condition (ECOG: 0–1) and had no serious comorbidities. The BRAF/MEK inhibitors (dabrafenib/trametinib) were administered daily with maximal recommended doses of dabrafenib (150 mg, orally, twice daily) and trametinib (2 mg, orally, once a day). Patients without a BRAF mutation were treated with immunotherapy: nivolumab (240 mg per 2 weeks, or 480 mg per 4 weeks), pembrolizumab (200 mg per 3 weeks, or 400 mg per 6 weeks), or nivolumab + ipilimumab (1 mg/kg and 3 mg/kg, respectively per 3 weeks). Patients with brain metastasis detected by computed tomography (CT) or magnetic resonance (MR) imaging were treated with surgery and/or radiotherapy, depending on the clinical indication, before the systemic treatment. A physical examination was performed after each dose of systemic treatment, and a blood test at least every 12 weeks. CT and/or MR were performed at least every 12 weeks of treatment. Clinical outcome was assessed using TTP calculated from the date of initiation of the systemic therapy until progression documented by CT according to the *Response Evaluation Criteria in Solid Tumors* (RECIST, ver 1.1) or clinical examination. All patients provided written informed consent, prior to participation. The study was approved by the Ethics Committee of the NRIO, Gliwice Branch (approval no. KB/430-86/19), and complied with the Declaration of Helsinki. The demographic and clinical characteristics of the patients are presented in Table 1. NGS analysis was performed on samples derived from 14 patients with the highest cfDNA concentration. 

### 2.2. Blood Sample Collection, DNA Isolation, and FFPE Samples 

Blood samples (approximately 8 mL) were collected in BD Vacutainer^®^ PPT™ K2E tubes (cat. No. 362799) and centrifuged within one hour at 1100× *g* for 10 min. The plasma was additionally centrifuged at 16,000× *g* for 10 min, transferred to cryogenic vials, and stored at −80 °C until processed. The cfDNA was isolated from 3–4 mL of patients’ plasma (liquid biopsy, LB) using a QIAamp Circulating Nucleic Acid Kit (Qiagen, Hilden, Germany) according to the manufacturer’s protocol, with some modifications (without carrier RNA). The quantity of the DNA was measured using a Qubit™ dsDNA HS Assay Kit and Qubit 3.0 Fluorimeter (Thermo Fisher Scientific, Waltham, MA, USA). When required, the cfDNA was concentrated using a standard protocol. Briefly, 2.5 volume of 100% ETOH, 1/10 volume of 3 M sodium acetate (pH 5.2), and 1 µg of glycogen were added to the samples, and DNA was precipitated at −20 °C for 72 h. After centrifugation and washing with 80% EtOH, the cfDNA was dissolved in an appropriate amount of water. For NGS analysis of tumor DNA, archival DNA previously isolated from formalin-fixed, paraffin-embedded (FFPE) samples for BRAF diagnostic were used. It was isolated using a Maxwell RSC FFPE Plus DNA kit (Promega) using a Maxwell RSC machine (Promega), according to the manufacturer’s protocol. The quantity was measured as described above. Most melanoma samples contained more than 50% tumor cells and the median tumor cell percentage was 77.5%. 

### 2.3. QPCR 

BRAF V600 mutations were analyzed by quantitative real-time PCR (qPCR) using an AmoyDx BRAF V600 Mutation Detection Kit (Amoy Diagnostics Co, Xiamen, China) on the QuantStudio 12 K Flex Real-time PCR system (Thermo Fisher Scientific, Waltham, MA, USA), according to the manufacturer’s protocol.

### 2.4. Sequencing of Tumor and Cell-Free DNA

A total of 30 samples (19 LB samples and 11 FFPE samples) derived from 14 patients were sequenced. Oncomine™ Pan-Cancer Cell-Free Assay (off-the-shelf panel that targets 272 amplicons within 52 known cancer genes) was used for library preparation using 2.5 to 20 ng of the tumor or cfDNA. The libraries were synthesized manually following the manufacturer’s protocol, quantified with an Ion Library TaqMan Quantitation Kit (Thermo Fisher Scientific, Waltham, MA, USA), diluted to 100 pM, and pooled for automated templating with an Ion 540™ kit for the IonChef Instrument. Sequencing was performed with the GeneStudio S5 system and Ion 540™ chips (4–6 LB samples/chip and 10 FFPE samples/chip). The average total mapped reads per sample was 5.2 mln for FFPE samples and 14.2 mln for LB samples with an average coverage of 15,298 for FFPE samples and 44,115 for LB samples. The median molecular coverage was 2842 for LB samples and 921 for FFPE samples. 

### 2.5. Data Analyses and Variant Selection 

Sequence data were processed using the Torrent Suite 5.16.0 pipeline software optimized for the Ion Torrent platform to perform raw data analysis, base calling, remove low-quality reads, and make alignments to the human genome (GRCh37/hg19). Variant calling was performed with Ion Reporter Server 5.16 and the software Oncomine TagSeq Pan-Cancer Liquid Biopsy—w2.4—Single Sample for LB samples, which, according to the manufacturer, detects and annotates low-frequency variants, including SNPs/InDels (down to 0.1% limit of detection), Fusions, and CNVs from targeted nucleic acid libraries (DNA and RNA) from the Ion Torrent Oncomine™ Pan-Cancer Cell-Free Assay. The FFPE samples were analyzed using Oncomine TagSeq Pan-Cancer Tumor—w2.4—Single Sample workflow, which, according to the manufacturer, detects and annotates low-frequency variants, including SNPs/InDels (down to 0.5% limit of detection), Fusions, and CNVs from targeted nucleic acid libraries (DNA and RNA) from the Ion Torrent Oncomine™ Pan-Cancer Cell-Free Assay. Apart from the default variant filter (Variant Matrix Summary), we performed also variant selection with our filter chain with the following parameters: allele frequency 0–100%; *p*-value < 0.05; UCSC Common SNPs not in; and variant type: CNV, Fusion, InDel, MNV, SNV, and Alternative Allele Count >5. For FFPE sample variants with an allele frequency <1%, an alternative allele molecular count <10, and variant type C/T or G/A were not included in the analysis. Selected variants were visually examined using .bam files generated with the Torrent Suite—FileExplorer v.5.16.0.0 plugin together with Alamut Visual v.2.11 software (Sophia Genetics, Boston, MA, USA). The data are openly available in FigShare at doi: 10.6084/m9.figshare.14994141.

### 2.6. Statistical Analysis 

Data were analyzed depending on data distribution assessed with the Shapiro–Wilk test. Spearman’s rank-order correlation coefficient was assessed to examine the correlation between variables. All analyses were performed using the R statistical software package version 4.0.1 released in June 2020 (R Foundation for Statistical Computing, Vienna, Austria, http://www.r-project.org, accessed on 5 January 2021). A two-sided *p*-value < 0.05 was considered statistically significant. 

## 3. Results 

### 3.1. cfDNA—Correlation Analysis 

Cell-free DNA was isolated from the plasma of 22 patients before first-line treatment. The concentration ranged from 4.6 to 705 ng/mL (median 9.13 ng/mL). All the patients with a cfDNA concentration above 50 ng/mL had metastasis in at least three localizations. The correlation analysis revealed that the cfDNA level correlated positively with the LDH concentration when all samples (*n* = 22) were compared as well as when only samples (*n* = 16) with elevated LDH levels (>220 U/L) were analyzed (Figure 1A,B). A negative correlation between the LDH level and time to progression after first-line therapy was also observed (Figure 1C). On the other hand, we did not observe any statistically significant correlation of the cfDNA level with progression-free survival in our cohort (rho = −0.0023, *p* = 0.99). For next-generation sequencing, 14 patients with the highest cfDNA concentration were selected. 

### 3.2. NGS Analysis of Tumors and cfDNA 

Targeted NGS (tNGS) analyses using Oncomine™ Pan-Cancer Cell-Free Assay were done in 30 samples of 14 patients. The number of analyses and type of DNA used for each patient is presented in Table 2. We used the Oncomine™ Pan-Cancer Cell-Free Assay for both liquid biopsy and FFPE samples; however, to limit the potential detection of false-positive variants introduced by formalin fixation in FFPE tumors, we applied a more restricted filter chain for variants selection in tumor DNA samples (see Materials and Methods). For all 14 patients, we compared the qPCR and NGS results of the *BRAF* mutation status in tumors and/or the LB of the same patient (Table 3). The analysis revealed 91% concordance both between the qPCR and NGS results in tumor samples (inter-platform concordance) and between tumor and plasma NGS analysis of the *BRAF* mutation status of the same patient (inter-tissue concordance). Variant allele frequency (VAF) of the *BRAF* mutation ranged from 18.7% to 50.7% in tumors and from 0.2% to 43.2% in cfDNA before therapy. An increase in the BRAFV600E frequency was observed in consecutive samples in three out of five *BRAF*-positive patients who had two LB samples analyzed (before the first and second therapy). 

In addition to the *BRAF* mutation, we have identified pathogenic variants in 14 other genes (*TP53*, *NRAS*, *CTNNB1*, *MAP2K1*, *RAF1*, *FGFR3*, *CCND1*, *MAP2K2*, *PIK3CA*, *PTEN*, *GNAS*, *ERBB3*, *MTOR*, and *MET*) (Table 4, Figure 2). There were 14 SNV/MNV variants and 3 amplifications (*FGFR3*, *CCND1*, and *MET*) detected in 28 samples in total. Most of the patients (11; 78.6%) have at least one non-BRAF mutation detected in either tumor DNA or cfDNA (or both). One patient did not have any pathogenic mutations detected. *CTNNB1* was the second (after *BRAF*) most frequently mutated gene in our cohort (4 cases of variant S45F) followed by *TP53* (3 cases; variants: Q104*, C176F, R273C) and *NRAS* (two cases of variant Q61K and one case of the double variants Q61R and Q61K). The VAF of the variants ranged from values at the border of a limit of detection (LOD), as in the case of, e.g., *GNAS*^R201H^ (0.08%) and *PIK3CA*^G1049R^ (0.07%), to 47% for *TP53*^Q104^. The tumor-plasma concordance of the non-BRAF mutations was 28%. A total of 5 out of 18 variants were present in both tumor and liquid biopsy, 11 were detected only in tumor DNA, and 2 in cfDNA. In five patients, we did not detect any variants in their liquid biopsies despite the identification of mutations in tumor samples. On the other hand, NGS analysis of the tumor and two consecutive liquid biopsies in Patient #1 showed 100% concordance between all samples for three alterations (Table 3 and Table 4). Moreover, a significant increase in VAF in the consecutive LB samples was observed. 

### 3.3. Selected Mutations and Patient Case 

#### 3.3.1. CTNNB1 (β-Catenin) 

The *CTNNB1*^S45F^ mutation was detected in four patients (#6, #9, #10, and #12; 28%, all *BRAF* positive) with a 32.2%, 2.1%, 0.4%, and 0.7% alternative allele frequency in their tumor samples, respectively. In Patient #6, with the highest VAF, the mutation was also detected in the liquid biopsy with a frequency of 23.5%. In the remaining patients, this variant was not detected in cfDNA. Two patients with VAF above 1% had a significantly shorter time to progression during first-line therapy (16 and 31 weeks) as compared to patients with a low frequency of this variant (40 and 52 weeks, still on treatment). The patient with the highest VAF was treated with targeted therapy for 7 weeks. He responded to the treatment; however, due to severe adverse effects, the treatment was terminated and immunotherapy with nivolumab was administered for the subsequent 24 weeks. The second patient harboring this mutation (VAF—2.1%) was diagnosed with nodular pT3b melanoma and relapsed after 7 years with metastasis to the brain. He was treated with targeted therapy, which lasted 16 weeks. The patient experienced partial regression of brain tumors, but new metastasis appeared in the mediastinum. Two patients with a low fraction of *CTNNB1*^S45F^ had metastasis in one or two localizations and no secondary tumors in the brain. At the moment of manuscript submission, they have been treated with targeted therapy for at least 40 weeks. 

#### 3.3.2. TP53

Three various mutations of *TP53* were detected in three patients (all BRAF positive): Q1004* (#1), C176F (#2), and R273C (#7), with a VAF of 41.25% (tumor), 0.08% (liquid biopsy 1), and 35.8% (tumor), respectively. In two patients, the *TP53* mutation was detected in both the tumor and cfDNA, while in Patient #2 only in the liquid biopsy. Additionally, in Patient #1, the mutation was also identified in consecutive liquid biopsies after the first therapy. All patients had metastasis in at least three localizations and two of them progressed to the brain. They were treated with targeted therapy and responded with a stable disease (#1) and partial response (#2 and #7), which, however, did not last long (20 and 28 weeks, respectively). For Patient #1, the interesting tumor genetic profile found is described separately below. 

#### 3.3.3. Patient #1 

A 46-year-old woman was diagnosed with pT2b (10 mitoses/mm^2^) trunk melanoma (2016). After 3 years of regular control in an outpatient clinic, brain metastasis was detected. After resection and pathomorphological confirmation, stereotactic radiotherapy was performed. Further diagnostics (PET-CT) identified other metastases in the lymph nodes, lung, and peritoneum, and the patient was treated with anti-BRAF + anti-MEK therapy. She responded partially at some localizations and with the progression of other lesions. After 33 weeks, targeted treatment was terminated due to the disease progression and immunotherapy was administered. However, after 12 weeks, CT and MR examinations confirmed disease progression in the brain and other locations during immunotherapy. Two weeks later the patient died. Genetic analysis of the brain metastasis and two consecutive liquid biopsies (before the first- and second-line therapy) revealed the presence of three main genetic alterations: *BRAF*^V600E^, *TP53*^Q104*^, and amplification of cyclin D1. They were present in a high frequency in the tumor DNA (brain metastasis) and liquid biopsy before treatment and increased during treatment, reaching a 93% and 35% allele frequency, respectively, and 5.17 copies of *CCND1*. Additionally, two new low-frequency mutations were identified (*MAP2K2*^Q60P^ and *GNAS*^R201H^). The results of the genetic analysis of this patient’s tumor and cfDNA are shown in Figure 3. 

## 4. Discussion 

The main aim of this study was to test the performance of Oncomine™ Pan-Cancer Cell-Free Assay and the HD technology of the Ion Torrent platform in NGS analysis of tumor and liquid biopsy melanoma samples. The analysis was performed on 30 samples derived from 14 advanced melanoma patients treated either with targeted therapy or immunotherapy. 

In general, the analysis confirmed that Ion Torrent HD technology is a sensitive and reliable technique that enables the identification of low-frequency mutations (>0.1%) in ccfDNA, as manifested by the high inter-platform concordance (>90%) as well as tumor-plasma concordance of the *BRAF* mutations (>90%). Discordant results for one tumor sample (out of 11 NGS-qPCR matched pairs analyzed) was most likely caused by the low technical quality of sequencing as the mean coverage and median molecular coverage of this sample was much below the average for all samples (993 and 54, respectively, vs. 15,230 and 921). Our results of inter-platform concordance of the *BRAF* status are in line with other reports [13]. The observed concordance for the tumor-plasma NGS *BRAF* genotyping was also similar to the tissue-plasma concordance reported for the more sensitive method, ddPCR [14], and are in line with the results obtained for actionable mutations in other cancers [15]. On the other hand, we noted relatively low tumor-plasma concordance for non-BRAF genetic alterations (28%) as compared with other studies [15], including the one performed in melanoma samples using the same platform (the Ion Torrent) [16]. The possible explanation is discussed below. 

The usage of dual molecular barcoding in Ion Torrent HD technology enabled the identification of very rare alterations (<0.1% of VAF) in the liquid biopsy samples. However, these low-frequency variants were identified only in samples with a high technical quality of sequencing (coverage >40,000 and molecular coverage >3500). Accordingly, in one sample of very low sequencing quality (#6 liquid biopsy; coverage—1182 and molecular coverage—12), only variants with a >20% VAF were identified. This suggests that meeting the manufacturer’s technical recommendations is required to reliably sequence liquid biopsy samples using dual-molecular barcoding. This sophisticated molecular technology proved useful for highly sensitive sequencing of diluted ctDNA [4]; however, it may raise doubts when sequencing tumor FFPE DNA with a relatively high number of G > A/C > T nucleotide modifications. This type of DNA damage is caused by formalin fixation and may even reach 20% of all genetic changes in FFPE samples [17]. To limit detection of such artifacts, we filtered out all variants with a VAF < 1%, alternative allele molecular count <10, and variant type C/T or G/A. On the other hand, these types of alterations are the most common genetic changes in melanoma as a result of UV-introduced pyrimidine dimers [18]. In a diagnostic routine, use of uracil-DNA glycosylase enzyme to reduce DNA-related artifacts is recommended [19]. Using a highly sensitive tool to sequence tumor DNA samples is questionable also for another reason. Namely, a high amount of tumor cells in biopsied material (in our project exceeding 50%) results in the detection of very rare variants in these samples, which are thus undetectable in liquid biopsy. That is why tumor-plasma concordance of non-BRAF mutations in our study was rather low (28%) and it concerned only variants with VAF in tumor samples higher than 30%. Furthermore, the amount of cfDNA taken for library synthesis may be critical for detection of variants in plasma. In our study, libraries of ten samples were prepared using less than 15 ng cfDNA. In only one of these samples (Patient #11), a mutated variant with relatively high VAF (8%) was identified. It seems that the required amount of cfDNA may be one of the limiting factors for using this technology in longitudinal liquid biopsy genetic profiling, as the median plasma concentration of cfDNA in advanced melanoma patients is approximately 10 ng/mL (our results and [20]). Another reason for the low tumor-plasma concordance in our study may be the too low VAF of the mutations identified in tumors for some genes (e.g., *MTOR*, *PIK3CA*, *ERBB3*, and *RAF1*), raising the question of whether they are driver mutations or just passenger variants. In turn, detection of these particular mutations stems from the gene panel used for NGS. We tested Oncomine™ Pan-Cancer, targeting 52 cancer-associated genes including several driver melanoma genes, while, for example, Diefenbach et al. tested a custom melanoma panel for which they observed 66.7% tissue-ctDNA positive concordance [16]. On the other hand, successful usage of Ion Torrent HD technology for longitudinal monitoring of genetic variants in blood was possible in the case of Patient #1. Sequencing of the tumor and two liquid biopsies of this patient revealed three main genetic alterations: *BRAF* and *TP53* mutations as well as *CCND1* amplification. Their VAF and CNV increased in consecutive ctDNA samples, suggesting a selection of melanoma clones harboring those genetic changes. This real-time liquid biopsy genetic profiling confirms the potential applicability of ctDNA sequencing and plasma sampling in clinical diagnostics as a companion tool for disease monitoring, early detection of actionable mutation, and clinical decision making. However, the financial aspect of this test should also be considered before implementation to the clinics, as the relatively high cost may be yet another limiting factor. 

Because of the small patient cohort, association analysis of the genetic alterations and clinical parameters was impossible. However, some of the mutated genes detected are worth discussing and further investigation. They are *CTNNB1* and *TP53*, two of the most frequently mutated genes in our cohort. Mutations in β-catenin are present in various cancers [21], including melanoma [22]. In our small cohort, we detected four cases of *CTNNB1*^S45F^, which is a much higher frequency (28%) than that observed in previous studies (7% [22] and 3% [23]). Interestingly, a higher frequency of the mutated allele was present in patients who progressed relatively early on both therapies, when compared to patients with a low frequency. Though the difference in duration of the therapy may stem from various stages of the disease of the patients, the influence of this mutation cannot be excluded. Previous studies suggested the contribution of this genetic alteration to resistance to targeted therapy [24] as well as immunotherapy [25]. The potential mechanism of melanoma insensitivity to treatment may involve resistance of the mutated cells to apoptosis [26]. A *TP53* mutation may also impair a response to targeted therapy [27] and immunotherapy [28]. In our small cohort, we have identified three various pathogenic mutations of this cancer suppressor. The patients harboring these alterations responded to targeted therapy with disease stabilization or partial response, which lasted shorter than the median TTP observed for this treatment. Second-line immunotherapy administered to two patients also did not prolong patients’ lives significantly. These observations together with previous reports justify further research on the role of TP53 in melanoma resistance to therapies.

## 5. Conclusions

Our results confirmed both the potential as well as limitations of liquid biopsy genetic profiling, as previously addressed by other researchers. HD technology and the Ion Torrent platform seem to be reliable tools for targeted sequencing of liquid biopsy samples. They enabled the identification of variants with various VAF ranges and confirmed its applicability for future studies on cfDNA genetic profiling in melanoma. However, caution should be taken when applying this technology in the sequencing of low-concentration cfDNA samples as well as melanoma FFPE samples, especially for tumor-plasma concordance studies. A custom melanoma-specific panel could be more applicable for genetic profiling of melanoma than Oncomine™ Pan-Cancer Cell-Free Assay. 

## Figures and Tables

**Figure 1 genes-12-01080-f001:**
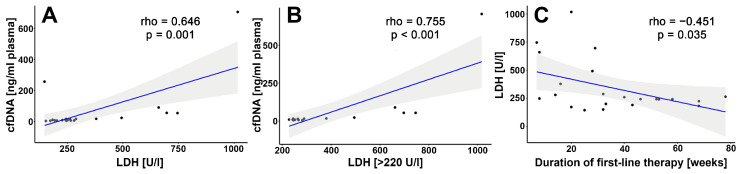
Spearman’s rank-order correlation analysis of: (**A**) LDH level and cfDNA concentration (*n* = 22); (**B**) elevated LDH level (>220 U/L) and cfDNA concentration (*n* = 16); (**C**) LDH level and duration of first-line therapy (*n* = 22). *p*-values < 0.05 were considered statistically significant.

**Figure 2 genes-12-01080-f002:**
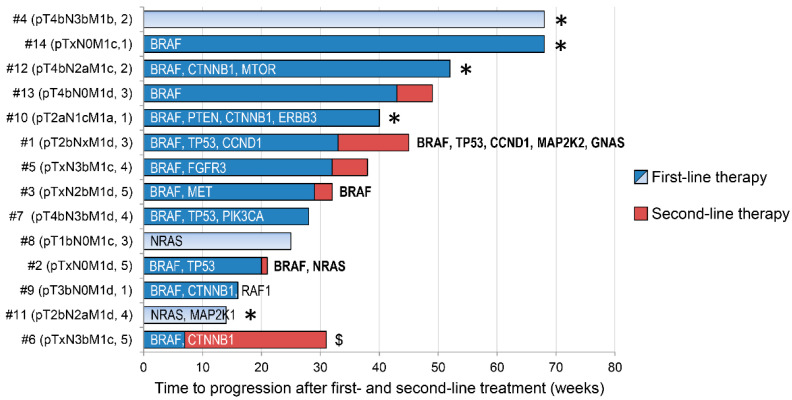
Summary of the mutated genes identified by targeted NGS in the context of clinical data. The patients are ordered according to the duration of the first-line therapy. Light blue and red bars denote immunotherapy, dark blue—targeted therapy. Genes with mutations detected before the second-line therapy (LB2 in three out of five patients: #1, #2, #3) are shown in bold. The numbers in parentheses behind the TNM stage represent the number of localizations with metastases. An asterisk denotes continuation of the therapy; $—this patient had discontinuation of first-line therapy due to adverse effects.

**Figure 3 genes-12-01080-f003:**
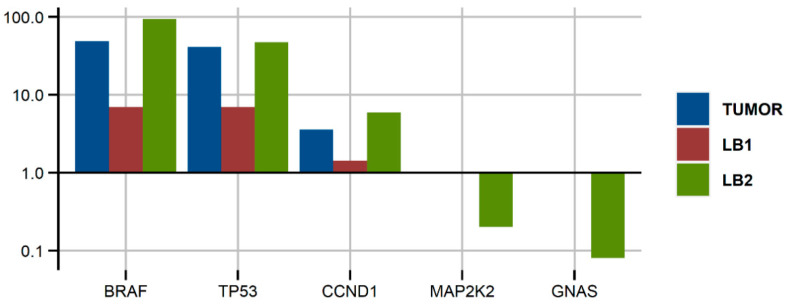
Variant allele frequency of the mutations (for *BRAF*, *TP53*, *MAP2K2*, and *GNAS*) and copy number variations ratio (for *CCND1*) detected in various samples of Patient #1; LB—liquid biopsy.

**Table 1 genes-12-01080-t001:** Characteristics of the patients.

Variable	Group	No. of Patients (*n* = 22)	%
Age, years	Median	52	
Sex	Female	13	60
	Male	9	40
Location of the primary tumor	Head and neck	4	18
	Trunk	10	45
	Extremities	5	23
	Lack of data	3	14
TNM Stage IV	M1a	3	14
	M1b	4	18
	M1c	8	36
	M1d	7	32
Therapy	Dabrafenib, trametinib	15	68
	Immunotherapy	7	32
Best response	CR	0	0
	PR	15	68
	SD	3	14
	PD	4	18
LDH at the start of therapy	Elevated	16	73
	Normal	6	27
Cell-free DNA (cfDNA) at the start of therapy (above or below median)	<10.3 ng/mL	11	50
	>10.3 ng/ml	11	50

**Table 2 genes-12-01080-t002:** Samples for NGS analysis.

DNA Sample	No. of Samples Per Patient	No. of Patients (*n* = 14)
LB1 ^1^	1	2
LB1 and LB2 ^2^	2	1
LB1 and T ^3^	2	7
LB1, LB2 and T	3	4

^1^ LB1—liquid biopsy before first-line therapy; ^2^ LB2—before second-line therapy; ^3^ T—biopsy of the tumor.

**Table 3 genes-12-01080-t003:** *BRAF* concordance analysis.

Patient No		Tumor			Liquid Biopsy 1		Liquid Biopsy 2	
	P/M ^1^	qPCR	NGS	VAF ^2^ %	NGS	VAF %	NGS	VAF %
1	M	V600	V600E	48.6	V600E	7.0	V600E	93.0
2	M	V600	V600E	18.7	V600E	25.0	V600E	35.0
3	M	V600	V600E	37.5	V600E	2.5	V600E	5.2
4	M	WT ^3^	WT	0.0	WT	0.0	WT	0.0
5	M	V600	NA ^4^	-	V600E	35.0	WT	0.0
6	P	V600	V600E	43.9	V600E	43.2	NA	
7	M	V600	V600E	31.4	V600E	0.7		
8	P	WT	WT	0.0	WT	0.0		
9	P	V600	WT *	0.0	V600E	0.3		
10	P	V600	V600E	50.7	V600E	16.8		
11	P	WT	WT	0.0	WT	0.0		
12	P	V600	V600R	33.0	V600R	0.2		
13	M	V600	NA	-	V600E	6.4		
14	P	V600	NA	-	WT *	0.0		

^1^ P/M—primary tumor/metastasis; ^2^ VAF—variant allele frequency; ^3^ WT—wild type; ^4^ NA—not analyzed; discordant results are marked with an asterisk.

**Table 4 genes-12-01080-t004:** Non-BRAF variants detected by targeted NGS in patient samples.

Patient No			Tumor	LB1	LB2
	GENE	MUTATION	VAF %	VAF %	VAF %
1	*TP53*	Q104 *	41.25	7.0	47.3
	*CCND1*	GAIN	3.58	1.43	5.9
	*MAP2K2*	Q60P	-	-	0.22
	*GNAS*	R201H	-	-	0.08
2	*TP53*	C176F	-	0.08	-
	*NRAS*	Q61R	-	-	0.7
	*NRAS*	Q61K	-	-	0.3
3	*MET*	GAIN	1.3	-	-
4	no variants		-	-	-
5	*FGFR3*	GAIN	NA	1.26	-
6	*CTNNB1*	S45F	32.2	23.5	NA
7	*TP53*	R273C	35.8	1.1	NA
	*PIK3CA*	G1049R	-	0.07	NA
8	*NRAS*	Q61K	24.0	-	NA
9	*CTNNB1*	S45F	2.1	-	NA
	*RAF1*	S257L	1.3	-	NA
10	*PTEN*	R173C	3.7	-	NA
	*ERBB3*	T355I	0.5	-	NA
	*CTNNB1*	S45F	0.4	-	NA
11	*NRAS*	Q61K	40.0	8.0	NA
	*MAP2K1*	P124S	0.92	-	NA
12	*CTNNB1*	S45F	0.7	-	NA NA
	*MTOR*	A1459P	0.56	-	
13	no variants		NA	-	NA
14	no variants		NA	-	NA

NA—not analyzed; discordant results are marked with an asterisk.

## Data Availability

The data presented in this study are openly available in FigShare at doi: 10.6084/m9.figshare.14994141.

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
