# Peer review of "NGS Analysis of Liquid Biopsy (LB) and Formalin-Fixed Paraffin-Embedded (FFPE) Melanoma Samples Using Oncomine™ Pan-Cancer Cell-Free Assay"

_genes, 2021, doi:10.3390/genes12071080_

Round 1
Reviewer 1 Report
This is a very well written paper, which presents one of the most advanced NGS techniques applied to patients suffering from advanced melanoma and with particular attention to cfDNA which shows a certain clinical relevance with respect to the outcome itself. The Materials and Methods section is well represented, just like the Results and Discussion. I just feel like recommending a very slight correction of some typos. Congratulations to the authors.
Author Response
Response to Reviewer’s comments
We are very grateful for reviewing our manuscript.
The typos and other editorial mistakes have been corrected.
Fo the authors,
Magdalena Olbryt
Reviewer 2 Report
In this manuscript, Olbryt and coworkers tested DNA from FFPE melanoma tissues and matched plasma samples with the commercially available ‘Oncomine Pan-Cancer Cell-Free Assay’ NGS kit to detect mutations and copy number alterations in multiple known cancer genes (N=52) and to perform concordance studies by comparing tissues and blood. To this aim, they extracted and quantified circulating DNA from 22 Stage IV melanoma patients. They found a positive correlation between cfDNA level and LDH concentration, and a negative correlation between time to progression after first-line therapy and LDH concentration. Only 14 out of 22 patients have their cfDNA sequenced, along with 11 matched DNA samples from FFPE tissues. Authors found concordance of 91% between tissue DNA tested with QPCR and NGS and between NGS results in matched tissue and plasma samples. Sequence analysis revealed that, apart BRAF gene, other 14 genes resulted mutated, in particular CTNNB1 and TP53. In 3 tumors and matched plasma no gene variants were detected by using this assay. In 5 out of 14 patients studied, gene variants different from BRAF variants detected in tumors could not be detected in plasma.
The here presented data are interesting, although some of them are not so original (e.g. LDH is a recognized negative prognostic factor in metastatic melanoma). The manuscript is too descriptive and, in some points, confused. In my opinion, the limits of the technology tested should be better described and discussed in the context of liquid biopsy application (identification and detection of circulating gene variants and their longitudinal monitoring in blood). In addition, the number of samples tested is too small and heterogenous to draw conclusions about the implication of described gene variants in resistance to therapy.
Minor issues:
QPCR is not described in Materials and Methods section.
Extraction and quantification of DNA from FFPE melanoma samples is not described in Materials and Methods section.
Author Response
Response to Reviewer’s comments:
Thank you for your valuable review of our manuscript. In the corrected version of the manuscript, we have tried to address all your remarks and suggestions as described below.
- The description of the results was corrected to be more concise and clear.
- The Discussion was supplemented with the suggested points and the conclusions adjusted to the results.
- The Methods were completed with a description of QPCR and DNA isolation from FFPE samples.
- The English spelling mistakes were corrected.
For the authors,
Magdalena Olbryt
Round 2
Reviewer 2 Report
The authors have addressed satisfactorily the questions raised by this reviewer. The manuscript has been substantially improved.
Author Response
Dear Reviewer,
Thank you for reviewing our manuscript.
For the authors,
Magdalena Olbryt